# Molecular Activation of the Kv11.1 Channel Reprograms EMT in Colon Cancer by Inhibiting TGFβ Signaling via Activation of Calcineurin

**DOI:** 10.3390/cancers13236025

**Published:** 2021-11-30

**Authors:** Najmeh Eskandari, Vitalyi Senyuk, Jennifer Moore, Zane Kalik, Qiyue Luan, Ian Papautsky, Arfa Moshiri, Maurizio Bocchetta, Seyed Alireza Salami, Shahrbanoo Oryan, Saverio Gentile

**Affiliations:** 1Division of Hematology Oncology, Department of Medicine, University of Illinois at Chicago, Chicago, IL 60612, USA; neskan3@uic.edu (N.E.); vsenyuk@uic.edu (V.S.); jmoore58@uic.edu (J.M.); zkalik@uic.edu (Z.K.); 2Department of Animal Sciences, Faculty of Biological Sciences, Kharazmi University, Tehran 15719-14911, Iran; 3Department of Bioengineering, University of Illinois at Chicago, Chicago, IL 60612, USA; qluan4@uic.edu (Q.L.); papauts@uic.edu (I.P.); 4Gastroenterology and Liver Diseases Research Center, Research Institute for Gastroenterology and Liver Diseases, Shahid Beheshti University of Medical Sciences, Tehran 19857-17413, Iran; a.moshiri@pasteur.ac.ir; 5Microbiology Research Center, Pasteur Institute of Iran, Tehran 13169-43551, Iran; 6Cardinal Bernardin Cancer Center, Department of Cancer Biology, Loyola University Chicago, Maywood, IL 60153, USA; mbocche@luc.edu; 7Department of Biotechnology, Faculty of Agricultural Science and Engineering, University of Tehran, Karaj 31587-77871, Iran; asalami@ut.ac.ir

**Keywords:** potassium channels, colon cancer, TGFβ-signaling, migration, calcineurin

## Abstract

**Simple Summary:**

In metastasis, cancer cells migrate away from the tissue where they first were generated and spread in other body compartments. During this process, stationary epithelial cells undergo multiple biochemical changes to become mesenchymal cells that present enhanced migratory abilities. Among numerous factors controlling the mesenchymal phenotype, TGFβ is one of the most important. Nevertheless, little is known about the mechanisms inhibiting the TGFβ biochemical pathways and metastasis remain the main reason of cancer mortality. We discovered that the protein called Kv11.1 potassium channel plays a major role in motility of colon cancer cells. We found that stimulating Kv11.1 activity with specific activator molecules produces a shutdown of the TGFβ pathway at its early stages. Consequently, cancer cell motility is suppressed by a reprogramming of the mesenchymal into epithelial phenotype. This research opens the possibility of using Kv11.1 activator molecules as a potential therapeutic strategy against metastatic colon cancer.

**Abstract:**

Control of ionic gradients is critical to maintain cellular homeostasis in both physiological and pathological conditions, but the role of ion channels in cancer cells has not been studied thoroughly. In this work we demonstrated that activity of the Kv11.1 potassium channel plays a vital role in controlling the migration of colon cancer cells by reversing the epithelial-to-mesenchymal transition (EMT) into the mesenchymal-to-epithelial transition (MET). We discovered that pharmacological stimulation of the Kv11.1 channel with the activator molecule NS1643 produces a strong inhibition of colon cancer cell motility. In agreement with the reversal of EMT, NS1643 treatment leads to a depletion of mesenchymal markers such as SNAIL1, SLUG, TWIST, ZEB, N-cadherin, and c-Myc, while the epithelial marker E-cadherin was strongly upregulated. Investigating the mechanism linking Kv11.1 activity to reversal of EMT into MET revealed that stimulation of Kv11.1 produced a strong and fast inhibition of the TGFβ signaling. Application of NS1643 resulted in de-phosphorylation of the TGFβ downstream effectors R-SMADs by activation of the serine/threonine phosphatase PP2B (calcineurin). Consistent with the role of TGFβ in controlling cancer stemness, NS1643 also produced a strong inhibition of NANOG, SOX2, and OCT4 while arresting the cell cycle in G0/G1. Our data demonstrate that activation of the Kv11.1 channel reprograms EMT into MET by inhibiting TGFβ signaling, which results in inhibition of motility in colon cancer cells.

## 1. Introduction

Targeted therapy is a novel and successful strategy in the treatment of colon cancer [1]. Ion channels belong to a large group of proteins that control a variety of biochemical signaling. Therefore, characterizing the role of a specific ion channel in cancer biology could help to develop a novel targeted approach against cancer. The Kv11.1 potassium channel is a surface membrane protein that is normally expressed in the human brain, heart, and colon tissues. In these organs, Kv11.1 activity plays a vital role in controlling electrical excitability. Recently, several studies have demonstrated that Kv11.1 can be aberrantly regulated in a variety of cancers, including carcinoma of the breast, pancreas, colon, and leukemia [2,3,4,5]. In these cells, changes in the Kv11.1 current activity relate to alteration of a variety of hallmarks of cancer, ranging from proliferation to metastasis; therefore, this channel could be considered as a potential target for cancer therapeutics [6,7,8,9]. Although blockade of the Kv11.1 channel negatively affects cancer growth [10], the use of Kv11.1 blocker molecules is discouraged for cancer therapy due to the high risk of severe side effects linked to Kv11.1 function loss, including ventricular fibrillation [11]. In contrast, in our previous work, we have demonstrated that stimulation of Kv11.1 activity with activator molecules did not produce significant side effects while altering biochemical pathways in cancer cells [12].

Stationary epithelial cells undergo multiple biochemical changes that enable the epithelial-to-mesenchymal (EMT) transition, which enhances the migratory abilities of cells [13]. EMT is finely tuned by a variety of components, including membrane receptors and transcription factors, that control the expression and/or repression of adhesive molecules. In physiological conditions, the EMT process can be eventually reversed into mesenchymal-to-epithelial (EMT → MET) transition and cells reacquire an epithelial phenotype to conclude critical processes during embryonic development or tissue regeneration. In contrast, in cancer cells EMT is not complete and cells maintain an enhanced migratory behavior, which, in turn, contributes to metastasis [14]. 

The cytokine transforming growth factor beta (TGFβ) is a major inducer of EMT in a variety of cancers, including carcinomas of the colon [15]. The TGFβ response starts when the surface membrane TGFβ receptor I (TFGBRI) and the TGFβ receptor II (TGFBRII) form a complex upon TGFβ ligand binding (TGFβ1, TGFβ2, TGFβ3) [16]. Consequently, activated TGFBRI phosphorylates the cytoplasmic receptors SMAD2 and SMAD3 (R-SMADs) on the conserved carboxy-terminal SSXS motif, which promotes R-SMADs to bind the transcription factor SMAD4. Then, the complex translocates into the nucleus and induces expression of several mesenchymal markers, including the SNAILs (SNAIL1 and SLUG) and ZEB families of transcription factors that transcribe for N-cadherin and suppress expression of the epithelial marker E-cadherin [17,18]. In addition, TGFβ is a potent regulator of cancer cell stemness [19]. Thus, TGFβ signaling is considered one of the most potent factors in tumor progression. In contrast, little is known about the pathways that inhibit TGFβ signaling and developing therapeutic approaches aiming to suppress TGFβ-dependent metastasis is very challenging.

TGFβ signaling strongly relies on reversible phosphorylation. Therefore, identification of a ser/thr phosphatase that controls TGFβ signaling offers the opportunity to better understand EMT and the metastatic cascade. Human cells present at least 11 genes encoding for protein phosphatase (PP) targeting phosphorylated serine or threonine (ser/thr) [20]. These enzymes can be organized into classes according to their specific role and/or to the activation pathway. For example, PP1 appears to be the most prevalent protein ser/thr phosphatase in mammalian cells and its abundant nuclear localization confers to this protein an important role in controlling transcription factors activity. In contrast, PP2B (calcineurin) and PP2C are mostly cytoplasmic and are rapidly activated by Ca^2+^ or Mg^2+^, respectively. It is clear that phosphatases can play a critical role in cancer [21,22] but the biochemical signaling in which they could be involved has not been studied thoroughly.

Interestingly, Kv11.1 has been found transiently expressed during development in the neural crest cells [23], suggesting that this channel might play a role in the EMT/MET cellular processes. In this work, we show that Kv11.1 activity controls colon cancer cell motility by affecting TGFβ-dependent signaling through activation of calcineurin. This event results in dephosphorylation of R-SMADs, inhibition of mesenchymal and stemness markers, and activation of epithelial factors. We conclude that stimulation of the Kv11.1 channel arrests migration of colon cancer cells by reversing EMT into MET.

## 2. Materials and Methods

### 2.1. Cell Culture, Antibodies, and Reagents

HT29 cells were cultured in Dulbecco’s modified Eagle’s medium (DMEM) (4.5 g/L glucose) (Corning, Manassas, VA, USA) with 1% GlutaMax (Gibco, Waltham, MA, USA), FET cells in DMEM/F12 (50:50) media (Corning, Manassas, VA, USA), and SW480 cells in DMEM media, supplemented with 10% fetal bovine serum (FBS) (Gemini Bio, West Sacramento, CA, USA) and 1% penicillin (10,000 units/mL)/streptomycin (10,000 µg/mL) (Gibco, Waltham, MA, USA) at 37 °C and 5% CO_2_. All antibodies were purchased from Cell Signaling Technologies, Inc (Boston, MA, USA). An anti-KCNH2 antibody was purchased from Santa Cruz (Dallas, Texas, USA) or Alomone Lab (Islrael). Recombinant human TGF-β1 and anti-hTGF-β RII FITC conjugated were purchased from R&D Systems (Minneapolis, MN, USA). E4031 and NS1643 were purchased from Alomone Labs (Jerusalem, Israel). Sanguinarine chloride was purchased from Tocris (Bristol, UK). Cyclosporine A was purchased from LC Laboratories (Woburn, MA, USA). 

### 2.2. Electrophysiology 

Potassium tail currents were recorded utilizing the gramicidin-perforated configuration of the patch clamp technique in voltage clamp mode (Section A.1).

### 2.3. RT-PCR Analysis and si-RNA

Total RNAs were extracted from cells using Ambion Trizol (by Life technologies, Carlsbad, CA, USA) (Section A.2). 

### 2.4. Wound-Healing Assay

Images were taken prior to addition of the drug (time 0), as well as after 24 h of the treatment using an inverted microscope (EVOS XL Core Imaging system, Life Technologies, USA) with 10× magnification (Section A.3).

### 2.5. Video Microscopy and Single Cell Tracking

Cells were seeded in glass bottom culture dishes at low density. Single cells were imaged an overnight (10-min interval) using the Vivaview auto imaging system (VivaView Incubator/Fluorescence Microscope, Olympus, Shinjuku City, Tokyo, Japan) at multiple positions per dish with 40× magnification. Manual cell tracking was performed for selected cells by using the Manual Tracking plugin in ImageJ. Wind-rose plots were produced with Chemotaxis and Migration Tool 2.0 (Ibidi GmbH, Gräfelfing, Bavaria, Germany).

### 2.6. Flow Cytometry Analysis 

After incubation, cells were washed and resuspended in the FACS buffer, and then they were sorted and analyzed on a flow cytometer (LSR Fortessa with HTS, BD Biosciences, San Jose, CA, USA) to determine the expression of TGFβ RII on the surface of the membrane (Section A.5). 

### 2.7. Cell Cycle Assay

Cells were harvested and washed twice with cold PBS. The pellets were resuspended in PBS with 1% formaldehyde and 0.2% Trition X-100. Then, the cells were incubated in PBS with DAPI for 2 h. The samples were analyzed with a flow cytometer (LSR Fortessa with HTS, BD Biosciences, San Jose, CA, USA) and Summit (Beckman Coulter Inc.; Fullerton, Brea, CA, USA). 

### 2.8. Cell Spheroid Formation and Drug Treatment on Agarose-Microwells

Cell spheroids were formed after a 2-day incubation and separated into the treatment and control groups. Following treatment, cell spheroids in both groups were immunofluorescently stained with E-cadherin antibody (primary antibody from Cell Signaling; secondary IgG antibody with AlexaFluor 488 from Abcam, Waltham, MA, USA). Fluorescent images were taken using a Zeiss 710 Laser Scanning Confocal Microscope (Zeiss, White Plains, NY, USA) (Section A.6).

### 2.9. Cell Apoptosis Assay

Cell apoptosis was determined by using a BD Pharmingen FITC Annexin V Apoptosis Detection Kit I (Section A.7).

### 2.10. Statistical Analysis

Data were expressed as the mean values ± standard error, and a *p*-value less than 0.05 was considered statistically significant (*t*-test). Representative experiments were replicated at least three times.

## 3. Results

### 3.1. Activation of the Kv11.1 Channel Inhibits Colon Cancer Cells Motility

To investigate the role of the Kv11.1 channel in colon cancer cell motility, we selected a panel of cells expressing the Kv11.1 channel (Figure 1A–C) and performed a wound-healing assay before and after treatment with Kv11.1 activator molecule NS1643 for 16 h (Figure 1D,E). Since motility in colon cancer cells has been associated with the activation of transforming growth factor TGFβ signaling, we used the SW480 colon cancer cell line, which lacks SMAD4 expression [24] as negative control. These experiments show that NS1643 significantly decreases the migratory ability of HT29 and FET cells, but no significant change was observed in SW480.

To study cancer cell migration in a quantitative manner at the single-cell level, we took advantage of a single-cell tracking method. Tracking analysis of HT29 or FET cells exposed to a Kv11.1 channel activator for up to 16 h demonstrated that NS1643 can significantly inhibit both the velocity and directionality of the cells (Figure 1F,G). The quantitative analysis showed that cells exposed to NS1643 presented a reduced velocity (HT29 median value of 0.23 µm/min in control cells versus 0.15 µm/min in NS1643-treated cells; FET median value of 0.19 µm/min in control cells versus 0.11 µm/min in NS1643-treated cells control cells) and shorter distance (HT29 median value of 224.2 µm in control cells versus 151.7 µm in NS1643-treated cells; FET median value of 191.2 µm in control cells versus 111 µm in NS1643-treated cells control cells).

Similar to the results from the wound-healing assay, activation of the Kv11.1 channel in SW480 presented a non-significant change in velocity (median value of 0.187 µm/min in control cells versus 0.13 µm/min in NS1643-treated cells) or distance (median value of 0.179.6 µm in control cells versus 144 µm/min in NS1643-treated cells control cells). These data demonstrate that pharmacological stimulation of the Kv11.1 channel activity produces a significant inhibition of colon cancer cell migration behavior and suggests that this effect could be mediated by inhibition of the TGFβ signaling.

### 3.2. Kv11.1 Activity Inhibits Mesenchymal and Stemness Phenotype in Colon Cancer Cells

Epithelial-to-mesenchymal transition (EMT) plays a key role in the acquisition of the migratory properties in cancer cells. The suppressing effects of Kv11.1 activator on colon cancer cell motility suggests that Kv11.1 activity may inhibit EMT in these cells. To evaluate this hypothesis, we monitored the expression level of several mesenchymal markers, including SNAIL1, SLUG, ZEB, TWIST, and c-Myc in HT29, FET, and SW480 cancer cells after treatment with NS1643. We found that NS1643 produced a significant reduction in all mesenchymal factors (Figure 2A,B). The proto-oncogene c-Myc is considered a central factor in determining the mesenchymal phenotype as well as in maintaining cancer stemness [25]. We monitored the effects of NS1643 on c-Myc activity, and we found that the treated cells, HT29 and FET, presented a strong reduction in c-Myc protein density (Figure 2B). In addition, the CDKNA, FASN, and GAD genes are normally repressed by c-Myc while cyclin D is activated. We found that cells treated with NS1643 presented a strong increased transcription of the CDKNA, FASN, and GAD genes (Figure 2C) and a down-regulation of cyclin D (Appendix A). In addition, NS1643 treatment produced a significant reduction in the expression of stemness indicators such as NANOG, OCT4, and SOX2 (Figure 2D) in HT29 and FET. Nevertheless, none of the mesenchymal or stemness markers was affected by NS1643 in the SW480 cell line. 

Furthermore, consistent with the role of SNAIL1 and SLUG transcription factors in promoting expression of N-cadherin (another key factor in the mesenchymal phenotype), we found that N-cadherin was significantly suppressed in the NS1643-treated cells (Figure 3A).

### 3.3. Kv11.1 Activator Stimulates Expression of the Epithelial Marker E-Cadherin in Colon Cancer Cells

The transcription factor ZEB is known to contribute to maintaining the mesenchymal phenotype by repressing the expression level of the epithelial marker E-cadherin. Based on the data showing that NS1643 inhibits ZEB expression we monitored the effect of the Kv11.1 channel activator on E-cadherin. We found that NS1643 treatments produced a significant increase in E-cadherin synthesis, as indicated by the augmentation at the mRNA (Figure 3B) and protein level (Figure 3C,D) in both HT29 and FET cells. However, no changes in E-cadherin expression were detected in the SW480 cell line.

Furthermore, we took advantage of a method of culturing cells in 3D by creating spheroids in which we tested the effect of NS1643 on E-cadherin expression. We found that these cell clusters strongly upregulated E-cadherin expression at the surface membrane after application of NS1643 (Figure 3E; Appendix A). The increase in E-cadherin occurred independently from the size of the spheroids and throughout the entire cell clusters, suggesting that NS1643 permeates the spheroids efficiently. 

Furthermore, as the increase in E-cadherin associates with cell–cell contact-dependent inhibition of proliferation, we monitored the effect of NS1643 on the cell cycle. We found that NS1643 treatment is associated with a strong inhibition of colon cancer cell proliferation as the cells arrested the cell cycle in the G0/G1 phase (Appendix A) without causing cell death (Appendix A).

### 3.4. Kv11.1 Activity Inhibits TGFβ Signaling

The TGFβ signaling is a major factor determining the mesenchymal characteristics in several cancers, including colon cancer. To examine whether the Kv11.1 activator affects TGFβ signaling we monitored phosphorylation of SMAD2/3 on the conserved carboxy-terminal SSXS motif in colon cancer cells that were incubated with NS1643 for different time periods. We found that these cells presented a significant and rapid decrease in SMAD2 phosphorylation in all three cell lines (Figure 4A–C). Nevertheless, suppression of Kv11.1 expression by si-RNA (Figure 4D) or pharmacologically induced (Appendix A) significantly abolished the NS1643-dependent dephosphorylation of SMADs and the related changes in EMT markers (Figure 4E), indicating that Kv11.1 is the direct target and mediator of NS1643.

To evaluate the specific action of TGFβ in our system, we serum-starved the cells and then added TGFβ1 alone or in combination with NS1643. Since all three TGFβ ligands present equal strength in activating the SMAD signaling we used only TGFβ1. Furthermore, to better understand the potency of TGFβ1, we tested its ability to phosphorylate R-SMADs as a function of time and dose (Appendix A). As expected, exposure to TGFβ1 produced significant and rapid phosphorylation of R-SMADs (Figure 5A). This event was followed by increased expression of the mesenchymal marker N-cadherin (Figure 5B). Nevertheless, application of NS1643 completely abolished the TGFβ1 effects on R-SMADs phosphorylation (Figure 5A) and inhibited expression of N-cadherin while augmenting E-cadherin (Figure 5B).

These data indicate that activation of the Kv11.1 channel rapidly and strongly inhibits TGFβ-dependent EMT by inhibiting R-SMADs phosphorylation. 

### 3.5. Kv11.1 Activation Inhibits TGFβ Signaling Via Calcineurin

To better understand the effects of NS1643 on TGFβ signaling we wanted to investigate the link between Kv11.1 activity and dephosphorylation of R-SMADs. Our data indicate that Kv11.1 activation produces a dephosphorylation of the SSXS domain in the R-SMADs carboxy-terminus. The expression level of TGFβR2, which presents the highest affinity for TGFβ1 ligand, or TGFBR1 also did not change in cells treated with NS1643 (Appendix A). These data suggest that NS1643-dependent dephosphorylation of R-SMADs is not determined by a lack of TGFβ receptors. Therefore, we tested the hypothesis that activation of Kv11.1 produces activation of a Ser/Thr protein phosphatase, which, in turn, dephosphorylates R-SMADs.

Remarkably, we found that application of NS1643 in combination with the Ca^2+^ and Mg^2+^ chelator EDTA strongly inhibited the NS1643-dependent dephosphorylation of R-SMADs (Figure 6A), suggesting that a divalent cation-dependent phosphatase (PP2B (calcineurin) that is activated by Ca^2+^ or PP2C that is activated by Mg^2+^) could control the Kv11.1-dependent dephosphorylation of R-SMADs.

To discriminate between calcineurin and PP2C, we took advantage of the alkaloid sanguinarine that has been characterized as a potent PP2C [26] inhibitor as well as promotes Ca^2+^ entry [27]. Remarkably, application of sanguinarine did not inhibit the ability of NS1643 to produce R-SMADs dephosphorylation (Figure 6B), suggesting that the Mg^2+^-dependent PP2C does not mediate this event. Application of sanguinarine alone or in combination with TGFβ1 also produced a strong dephosphorylation of R-SMADs (Appendix A), and application of EDTA inhibited the effect of sanguinarine on R-SMADs dephosphorylation (Appendix A). These data suggest a role for the Ca^2+^-dependent phosphatase calcineurin in mediating the inhibitory effects of NS1643 on TGFβ signaling. To further test the role of calcineurin in mediating the NS1643 effect on R-SMADs phosphorylation, we first treated the cells with the generic calcineurin inhibitor, cyclosporine A (CSA), alone or in combination with NS1643. Remarkably, we found that CSA alone produced a strong and rapid (5 min) increase in R-SMADs phosphorylation (Figure 6C), suggesting that inhibition of calcineurin reveals the basic activity of TGFbRII. Application of CSA also rescued the NS1643-dependent dephosphorylation of R-SMADs (Figure 6C; NS1643+CSA @ 15 min). Furthermore, while application of NS1643 produced the expected strong and rapid inhibition of TGFβ1-dependent SMAD2/3 phosphorylation (Figure 6C; NS1643 + TGFβ1), this event was completely abolished by inhibiting calcineurin (Figure 6C; NS1643 + TGFβ1 + CSA).

### 3.6. PP2B-γ Catalytic Isoform Mediates the Inhibitory Effect of Kv11.1 Activation on TGFβ Signaling

Calcineurin is a heterodimer complex between a regulatory and catalytic subunit. Mammalian cells present three genes, PPP3CA, PPP3CB, and PPP3CC, that encode, respectively, for calcineurin catalytic subunits, PP2Bα, PP2Bβ, and PP2Bγ [28]. To explore the clinical relevance of our findings, we conducted in silico analysis with the Human Protein Atlas database (www.proteinatlas.org, accessed on 9 October 2021) [29], which is a curated gene/protein expression database of publicly available microarray datasets that performs survival analyses based on selected biomarker expression levels. Analysis of the Kaplan–Meier plots for PPP3CA, PPP3CB, and PPP3CC genes revealed that high expression of PPP3CC associates with a better overall survival (OS) when compared with the cohort of patients with low expression (Figure 7A–D). In contrast, no significant difference was found for the PPP3CA or PPP3CB genes. These results suggested that PP2Bγ (encoded by PPP3CC) could be the relevant isoform that mediates the effects of Kv11.1 activation of TGFβ signaling. To test this hypothesis, we first suppressed PPP3CC gene expression in HT29 cells by si-RNA (Figure 7E). Western blot analysis using a pan-Calcineurin antibody revealed a significant reduction in the total PP2B protein density. Then we monitored the expression level of E-cadherin in these cells. We found that both R-SMADs phosphorylation and E-cadherin were strongly downregulated, suggesting that cells lacking PP2Bγ present an upregulated TGFβ signaling.

## 4. Discussion

Cancer metastasis is the spread of cancer cells to tissues and organs distant from the primary tumor and is the leading cause of cancer-related morbidity and lethality. The metastatic process is composed of a variety of progressive events among which the cellular biochemical pathways that control cell motility play a major role [30].

Alteration of several ion channels expression has been related to different aspects of cancer of different histogenesis but the role of a specific channel in cancer biology is still understudied. In this work, we discovered that the Kv11.1 channel can control colon cancer cell motility. By using qualitative and quantitative live cell imaging techniques we discovered that treatment of human-derived colon cancer cells with the small Kv11.1 activator molecule NS1643 strongly inhibited cell migration. This event is associated with downregulation of a variety of factors controlling the mesenchymal phenotype. Concurrently, these cells presented a strong upregulation of the epithelial marker E-cadherin. It has been well established that formation of the β-catenin/E-cadherin complex at the surface membrane are major factors of the stationary epithelial phenotype [31,32]. We have previously discovered that stimulation of the Kv11.1 channel in breast cancer suppresses the transcriptional activity of β-catenin by inhibiting its AKT-dependent nuclear translocation Wnt signaling [9]. Consequently, NS1643 promoted accumulation of β-catenin at the surface membrane. In the same work, we also discovered that NS1643 produced an increase in the E-cadherin protein (and membrane accumulation), but at the time, we did not understand the mechanism linking Kv11.1 activity to this event. In the current work we discovered that the NS1643-dependent increase in E-cadherin can be associated with a significant inhibition of the transcriptional activity of ZEB, TWIST, and SNAIL, which are known to suppress E-cadherin synthesis [33] and to be controlled by TGFβ signaling [34]. Therefore, by inhibiting the inhibitor (ZEB, TWIST, and SNAIL), NS1643 produces an increase in E-cadherin protein synthesis.

Acquisition of stemness behavior is a major contributor to cancer development [35]. Several EMT indicators are known to induce transcription of stemness markers. It has been demonstrated that TGFβ signaling can control cancer stemness via the ZEB1 transcription factor [36]. Our findings indicate that NS1643 produces downregulation of both EMT and stemness factors and arrests the cell cycle in the G0/G1 phase. These data indicate that Kv11.1 activity can control a potent anticancer mechanism that suppresses stemness behavior and the mesenchymal phenotype of cancer cells while promoting the stationary feature of the epithelial phenotype; a reversing process from EMT into MET. 

As TGFβ plays a major role in controlling EMT, we focused our attention on whether the Kv11.1 activity-dependent MET was mediated by an alteration in TGFβ signaling specifically. Remarkably, application of NS1643 effectively limited the ability of the TGFβ1 ligand to induce phosphorylation of the serine residues in the c-terminus of R-SMADs. Interestingly, the NS1643 reprogramming of cancer cells occurred only in HT29 and FET but not in SW480 cells, which are known to lack SMAD4, a major component of the TGFβ signaling, even though these cells express a functional Kv11.1 channel.

The inhibitory effects of NS1643 on TGFβ signaling could be potentially attributed to a variety of causes, including transcriptional, posttranslational events, or potential unspecific targets (e.g., the BK channel [37]).

The reduction in R-SMADs phosphorylation level after NS1643 application occurred already 10 min after the stimulation of Kv11.1 activity. Therefore, the rapidity of the NS1643-dependent effects suggests a significant change at the post-translational level rather than transcriptional. Protein phosphorylation is regulated by the competing activity of the protein kinase and phosphatases. We found that cells did not present any significant changes in the TGFβ receptor (kinase) expression even after 24 h of NS1643 treatment. This suggested that the potential candidate mediator of NS1643 effects on R-SMADs dephosphorylation was a ser/thr phosphatase. Remarkably, Kv11.1 silencing rescued the effects of NS1643 on both R-SMAD dephosphorylation and on inhibition of EMT markers synthesis, indicating that in the context of our colon cancer cell line system, the effects of NS1643 on TGFβ signaling was mainly mediated by Kv11.1. Nevertheless, we cannot completely exclude that other biochemical pathways controlling cancer cell motility could be affected by other possible targets of NS1643.

Interestingly, exposing cells to the divalent cation chelator EDTA, which makes Ca^2+^ and Mg^2+^ unavailable to cross the membrane, inhibited the NS1643-dependent dephosphorylation of the carboxy-terminal SSXS motif R-SMADs. This suggested a possible role for the divalent cation-dependent ser/thr phosphatases calcineurin (PP2B) or PP2C activity in mediating the effect of NS1643 on TGFβ signaling. To discriminate between the two ser/thr phosphatases, we took advantage of the dual nature of the natural alkaloid sanguinarine, which is a potent inhibitor of PP2C [26] and activator of Ca^2+^ entry [27]. Application of sanguinarine alone determined a reduction in the SMADs phosphorylation level. In addition, use of sanguinarine in combination with NS1643 did not rescue the effect of NS1643 alone on SMADs phosphorylation. These data suggested that the effects of NS1643 could be mediated by activation of Ca^2+^ entry rather than inhibition of PP2C. This hypothesis is sustained by our previous works in which we showed that stimulation of Kv11.1 activity produces Ca^2+^ entry into cancer cells [33]. This event can be explained by the fact that loss of K+ ions produces hyperpolarization, which provides a driving force for Ca^2+^ to enter the cell. Furthermore, long treatments with NS1643 (24 h) determined activation of Nuclear Factors of Activated T cells (NFATs)-dependent transcription, which was suppressed by inhibition of PP2B [34]. Therefore, in the current work we tested whether the Ca^2+^-activated phosphatase calcineurin would be the mediator of the rapid NS1643-dependnet effects on SMADs. In contrast with sanguinarine, application of the well-known calcineurin inhibitor, the immunosuppressant cyclosporine A alone, produced a significant increase in SMADs phosphorylation. This event can be explained by the basal TGFβRII kinase activity that prevails against the inhibited calcineurin. CSA in combination with NS1643 also strongly inhibited the ability of NS1643 to produce R-SMADS dephosphorylation, indicating that calcineurin mediates the inhibitory effects of NS1643 on TGFβ signaling. Therefore, we concluded that stimulation of the Kv11.1 channel activity produces hyperpolarization, which, in turn, drives Ca^2+^ entry through a not-yet identified Ca^2+^ channel. This Kv11.1 activity-dependent increase in intracellular Ca^2+^ can activate calcineurin rapidly (minutes), which produces dephosphorylation of R-SMADs. Consequently, transcription of mesenchymal markers is severely inhibited and cell motility is suppressed while expression of epithelial factor is activated and cell–cell contact is promoted. Nevertheless, we cannot exclude at this time that NFAT could participate to the described events; however, to address this topic properly, more specific experiments need to be performed, which will be the focus of another investigation.

Recently, increasing amounts of evidence have demonstrated that suppression of calcineurin activity is associated with carcinogenesis [38]. For example, use of cyclosporine A to prevent transplant rejection is often associated with increased risk of cancer [39]. However, to the best of our knowledge, this event has been only investigated prevalently in the context of NFAT signaling and the association between CSA and tumor growth has been explained by the lack of immune system activity to attack cancer cells. Our data show that cells in which the expression level of the PP2Bγ has been suppressed (si-RNA) present a reduced level of E-cadherin, suggesting that the PP2Bγ catalytic isoform of calcineurin controls the TGFβ signaling in colon cancer. Furthermore, patients expressing a high level of PPP3CC present a significantly better OS compared with patients with low expression of this subunit. These data endorse the clinical relevance of Kv11.1 activator molecules as potential therapeutic approach against cancer.

It has been understood that while Kv11.1 channel expression is deregulated in cancers of a different histology, its expression in a healthy human body is limited to few organs. Unfortunately, due to the high risk of severe side effects linked to Kv11.1 function loss, including ventricular fibrillation, the use of Kv11.1 blocker molecules is discouraged for cancer therapy. In contrast, in our previous work, we have demonstrated that stimulation of potassium channels with activator molecules did not produce significant side effects in mice [9,12], while altering the biochemical pathways to change metabolism inhibits tumor growth and metastasis in cancer cells [9,12,40,41,42,43]. Therefore, the use of a Kv11.1 activator as a means to stimulate calcineurin could be considered as a potential anticancer mechanism. 

Metastasis is an extraordinarily multifaceted process in which a variety of biochemical cascades underlying specific cellular events, including separation from the primary tumor, invasion through surrounding tissues, adhesion to basement membranes, and arrest in a distant target organ, must occur. Our work has demonstrated that pharmacological activation of Kv11.1 severely affects TGFβ signaling, which plays a critical role in cell motility. Interestingly, however, it also has been demonstrated that blockade (rather than activation) of Kv11.1 in colon cancer cells can inhibit metastasis [44]. Although it is not yet clear whether this event could be related to the Kv11.1 blocker-dependent inhibition of tumor growth [45], the study reveals the important finding that Kv11.1 can form macro complexes with proteins such as PI3K, AKT, and β-integrin, which play a major role in cell-matrix adhesion. Binding of β-integrin to the complex augmented the Kv11.1 current and, seemingly in contrast with our data [9], it favored tumor colonization to the liver. Although there is not yet evidence that this event produces changes in TGFβ signaling, it is clear that Kv11.1 (and probably other ion channels) appears to play a cross-cutting role as it can contribute to a variety of biochemical pathways that are not necessarily mutually exclusive or irreversible in controlling metastasis. Therefore, although more extended and specific investigations are needed to fully understand the role of Kv11.1 in metastasis, it is possible to speculate that each of these pathways rely on a specific electrochemical gradient and that changes in Kv11.1 activity (stimulation/inhibition) can determine dramatic alteration of these biochemical cascades, which can affect the metastatic process.

## 5. Conclusions

Our work indicates that stimulation of Kv11.1 activity produces Ca^2+^ entry, which, in turn, activates the ser/thr phosphatase calcineurin. Consequently, calcineurin dephosphorylates R-SMADs, resulting in inhibition of the TGFβ-dependent EMT, reversal of EMT into MET, and inhibition of migration. 

## Figures and Tables

**Figure 1 cancers-13-06025-f001:**
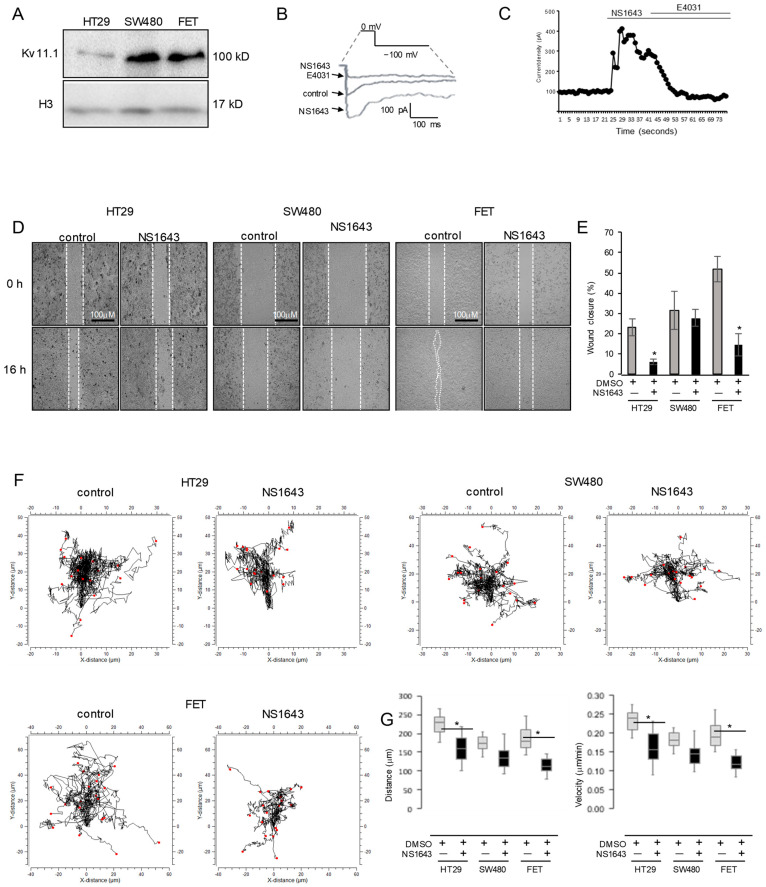
NS1643 inhibits migration. (**A**) Western blot analysis showing expression of Kv11.1 in HT29, SW480, and FET colon cancer cell lines. For the original Western blots, see Appendix A. (**B**) Representative Kv11.1 current in control HT29 cells in the presence of DMSO (control), NS1643 (activator; 50 µM) alone, or NS1643 + E4031 (blocker; 5 µM), with the voltage protocol shown. (**C**) Representative diary plot depicting the Kv11.1 peak currents (at −100 mV) as a function of time in HT29 cells before and after application of NS1643 alone or in combination with E4031. (**D**) Wound-healing assay with HT29, SW480, or FET cells treated with DMSO or 50 µM NS1643 for 16 h. (**E**) Quantification of the experiments in (**D**). Error bars indicate the standard error of the mean (*n* = 3; * *p* < 0.01). (**F**) Wind-rose plots for single-cell tracking experiments in HT29, SW480, and FET colon cancer cell lines treated with DMSO or 50 µM NS1643 for 16 h. (**G**) Box–whisker plots of the migration velocity and distance of the experiments in (**F**). *n* = 3; error bars indicate the minimum and the maximum values for the dataset; * *p* < 0.05.

**Figure 2 cancers-13-06025-f002:**
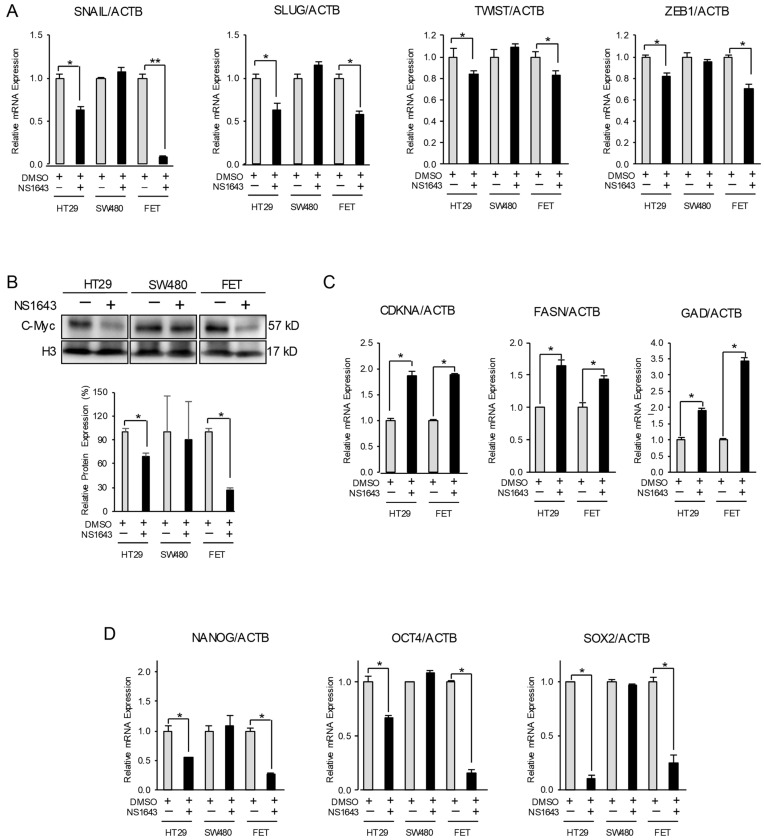
NS1643 inhibits expression of EMT markers. (**A**) Relative expression level of mRNA encoding for the EMT markers (SNAIL, SLUG, TWIST, and ZEB1) by RT-PCR in HT29, SW480, and FET cells treated with or without NS1643 (50 µM) for 48 h. Data = mean ± SEM; *n* = 3; * *p* < 0.01, ** *p* < 0.001. (**B**) Western blot analysis of c-Myc in HT29, SW480, and FET cells treated with or without NS1643 (50 µM) for 24 h. Data = mean ± SEM; *n* = 3; * *p* < 0.01. For the original Western blots, see Appendix A. (**C**) Expression level of mRNA encoding for CDKNA, FASN, and GAD (repressed by c-Myc) by RT-PCR in cells treated with or without NS1643 (50 µM) for 30 min in HT29 and 2 h in FET cells. Data = mean ± SEM; *n* = 3; * *p* < 0.001. (**D**) Expression level of mRNA encoding for the stemness indicators NANOG, OCT4, and SOX2 by RT-PCR in HT29, SW480, and FET cells treated with or without NS1643 (50 µM) for 48 h. Data = mean ± SEM; *n* = 3; * *p* < 0.001.

**Figure 3 cancers-13-06025-f003:**
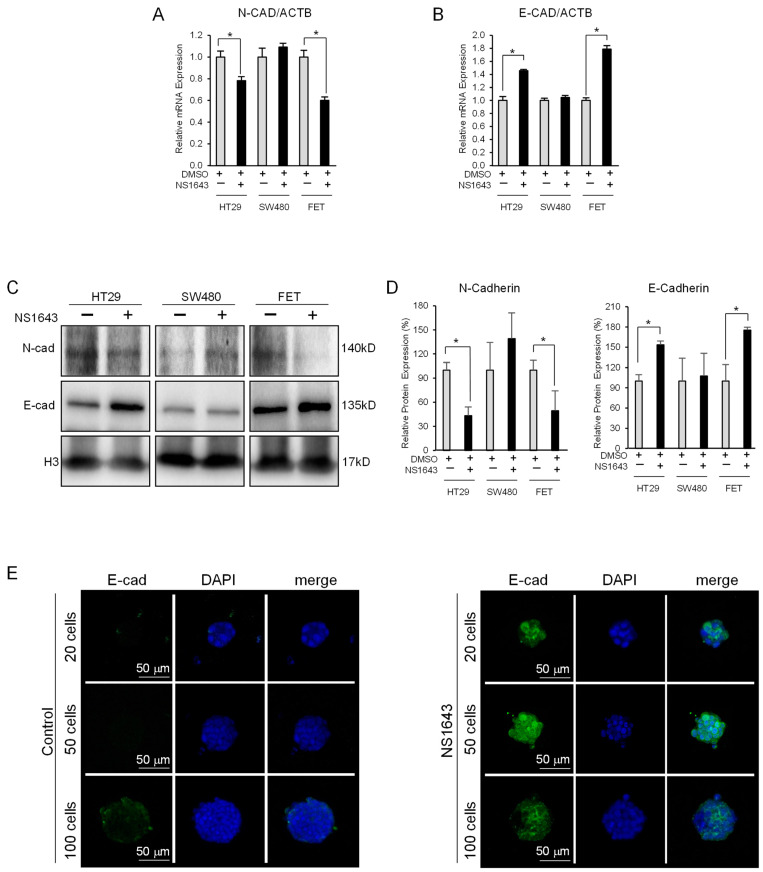
NS1643 inhibits expression of N-cadherin while augmenting E-cadherin (**A**) Relative expression level of mRNA encoding for N-Cadherin or (**B**) E-cadherin by RT-PCR in HT29, SW480, and FET cells treated with or without NS1643 (50 µM) for 48 h. Data = mean ± SEM; *n* = 3; * *p* < 0.01. (**C**) Western blot analysis showing expression of E-cadherin or N-cadherin in HT29, SW480, and FET colon cancer cell lines treated with or without NS1643 (50 µM) for 48 h. For the original Western blots, see Appendix A. (**D**) Quantification of the experiments in (**C**). Data = mean ± SEM; *n* = 3; * *p* < 0.001. (**E**) Representative confocal images of spheroids obtained from different HT29 cell-seeding densities (20 cells/well, 50 cells/well, and 100 cells/well) treated with DMSO (control) and NS1643 (50 µM) for 48 h.

**Figure 4 cancers-13-06025-f004:**
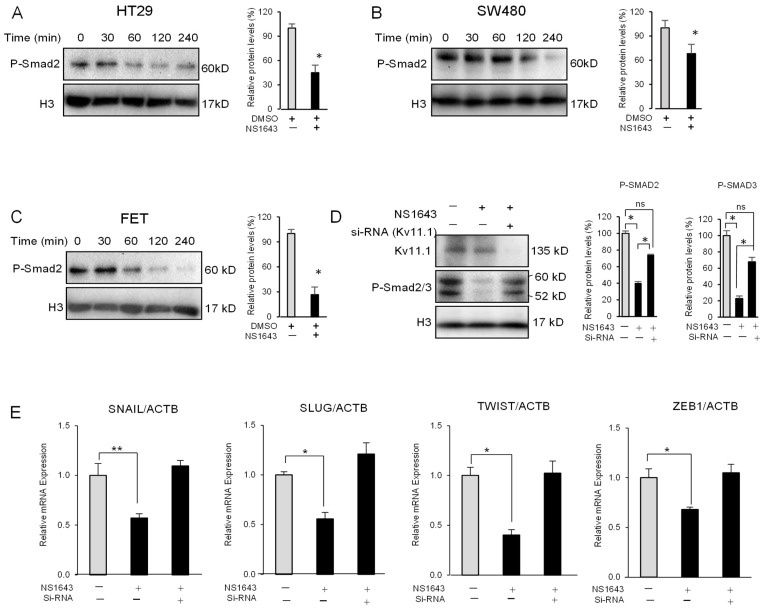
NS1643 inhibits R-SMADs phosphorylation (**A**) Western blot showing expression of phosphorylated SMAD2 (P-Smad2) in HT29, (**B**) SW480, or (**C**) FET colon cancer cell lines treated for different times, as indicated with NS1643 (50 µM). Cells were kept in full medium. Bar graph indicates quantification. Data = mean ± SEM; *n* = 3; * *p* < 0.05. (**D**) Western blots (Kv11.1 Ab Alomone Lab APC-109) showing the effect of NS1643 on SMADs phosphorylation in naïve HT29 cells or in cells in which Kv11.1 has been suppressed by si-RNA. ns: non-significant. Bar graph indicates quantification. Data = mean ± SEM; *n* = 3; * *p* < 0.01. (**E**) Relative expression level of mRNA encoding for the EMT markers (SNAIL, SLUG, TWIST, and ZEB1) by RT-PCR in naïve HT29 cells (light grey) or in cells in which Kv11.1 has been suppressed by si-RNA treated with or without NS1643 (50 µM) for 48 h. Bar graph indicates quantification. Data = mean ± SEM; *n* = 3; * *p* < 0.05, ** *p* < 0.01. For the original Western blots, see Appendix A.

**Figure 5 cancers-13-06025-f005:**
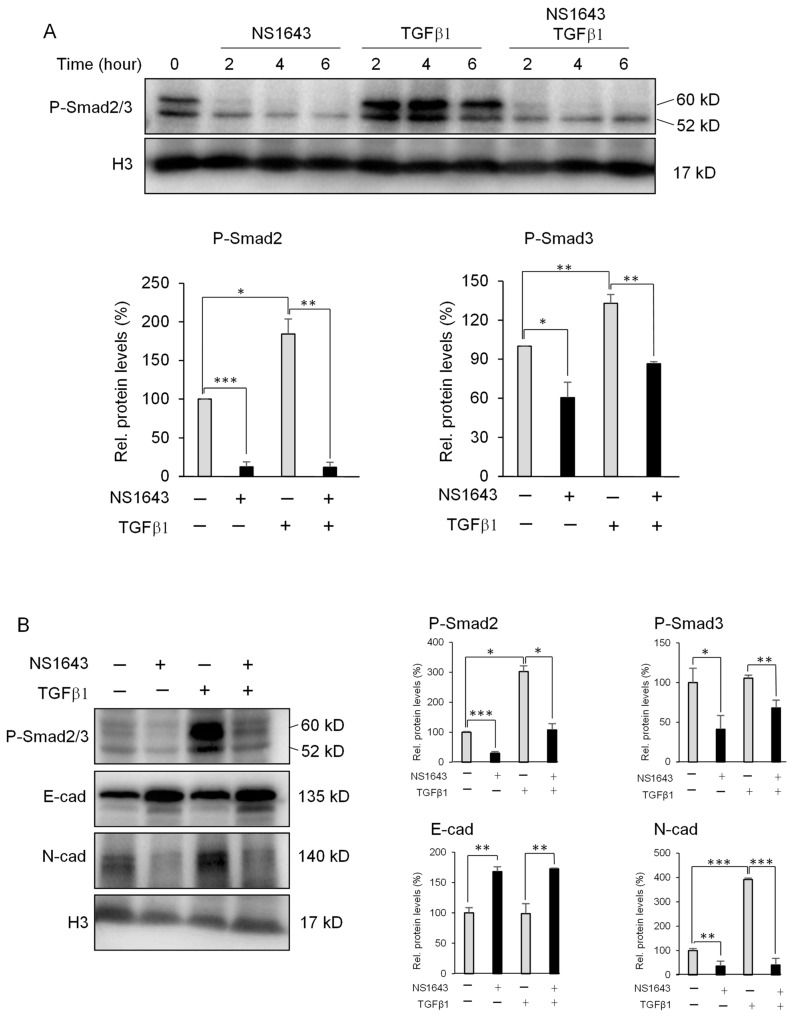
NS1643 abolishes the TGFβ1 effects on R-SMADs phosphorylation and EMT markers (**A**) Western blot showing expression of phosphorylated smad2/3 (P-Smad2/3) in HT29 cells treated for different times as indicated with NS1643 (50 µM) alone, TGFβ1 (2 ng/mL), or NS1643 and TGFβ1. Cells were in a serum-deprived medium for 16 h before application of drugs. Bar graphs indicate quantification of the experiments in (**A**) at the time point of 2 h. Data = mean ± SEM; *n* = 3; * *p* < 0.05; ** *p* < 0.01; *** *p* < 0.001. (**B**) Western blot analysis showing expression of phosphorylated SMAD2/3, E-cadherin, or N-cadherin in the HT29 cell line treated with NS1643 (50 µM; after serum starvation) for 48 h. Bar graphs indicate quantification of the experiments in (**B**). Data = mean ± SEM; *n* = 3; * *p* < 0.05, ** *p* < 0.01, *** *p* < 0.001. For the original Western blots, see Appendix A.

**Figure 6 cancers-13-06025-f006:**
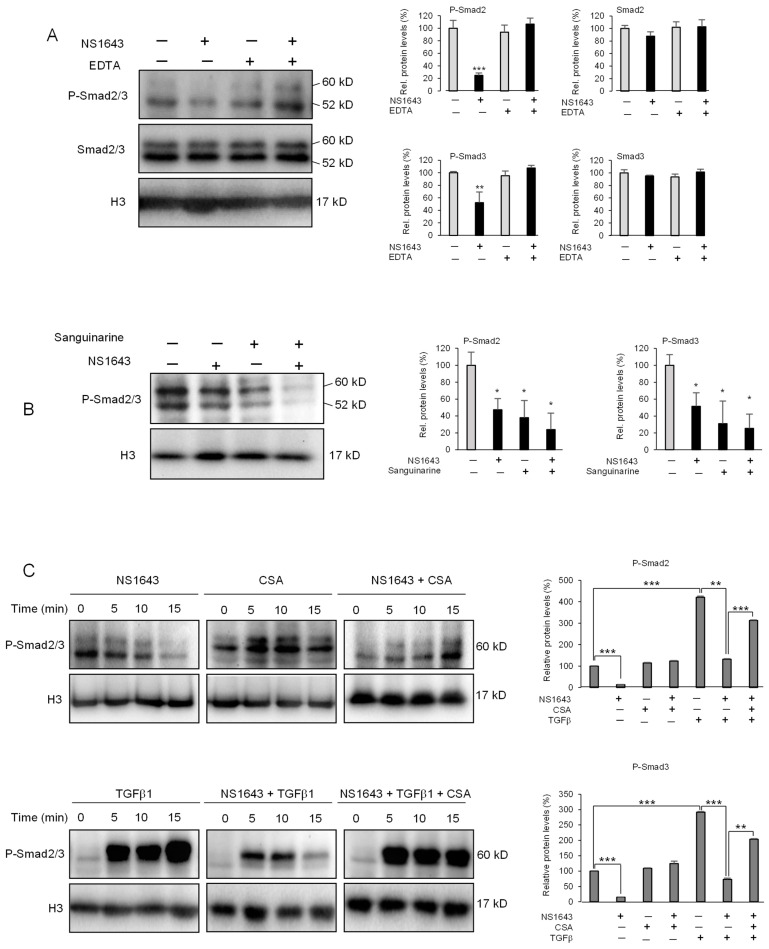
NS1643 inhibits TGFβ signaling via activation of calcineurin. (**A**) Western blots showing the expression level of phosphorylated smad2/3 and total smad2/3 in HT29 cells treated with DMSO, NS1643 (50 µM) alone, EDTA (10 mM) alone, or NS1643 + EDTA for 2 h. Bar graphs indicate quantification of the experiments in (**A**). Data = mean ± SEM; *n* = 3; ** *p* < 0.01; *** *p* < 0.001. (**B**) Western blot analysis showing expression of phosphorylated smad2/3 in HT29 cell line treated with DMSO, NS1643 (50 µM) alone, Sanguinarine (1 µM) alone, or NS1643 + Sanguinarine for 2 h. Bar graphs indicate quantification of the experiments in (**B**). Data = mean ± SEM; *n* = 3; * *p* < 0.05. (**C**) Western blot showing expression of phosphorylated smad2/3 in HT29 cells (serum-starved for 16 h) treated for different times, as indicated with NS1643 (50 µM) alone, Cyclosporine A (1 µM), TGFβ1 (2 ng/mL) alone, NS1643 + Cyclosporine A, NS1643 + TGFβ1, or NS1643 + Cyclosporine A + TGFβ1. Bar graphs indicate quantification of the experiments in (**C**). Data = mean ± SEM; *n* = 3; * *p* < 0.05, ** *p* < 0.01, *** *p* < 0.001). For the original Western blots, see Appendix A.

**Figure 7 cancers-13-06025-f007:**
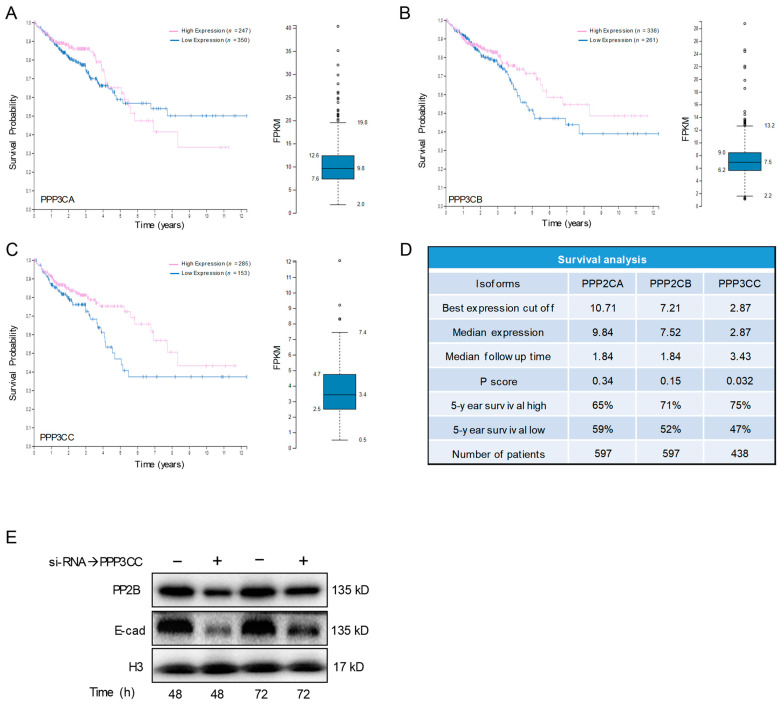
PP2B-γ Mediates the Inhibitory Effect of NS1643 on TGFβ Signaling. (**A**) Kaplan–Meier plots of overall survival in colon cancer patients with high (pink) and low (blue) expression of PPP3CA, (**B**) PPP3CB, or (**C**) PPP3CC. Side panel indicate the fragments per kilobase of transcript per million (FPKM) mapped reads. (**D**) Survival analysis of PPP3CA, PPP3CB, or PPP3CC gene expression in colorectal cancer patients. (**E**) Western blot showing the expression level of PP2B and E-cadherin at different time after that cells were transfected with si-RNA targeting PPP3CC. For the original Western blots, see Appendix A.

## Data Availability

The data presented in this study are publicly available.

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
