# Peer review of "Molecular Activation of the Kv11.1 Channel Reprograms EMT in Colon Cancer by Inhibiting TGFβ Signaling via Activation of Calcineurin"

_cancers, 2021, doi:10.3390/cancers13236025_

Round 1
Reviewer 1 Report
The authors partially addressed my major concern, regarding the specificity of NS1643 in the present study. They show now in Figure 4D that “Nevertheless, suppression of Kv11.1 expression by si-RNA (Fig. 4D) or pharmacologically induced (Supplementary Fig. 2B), completely abolished the NS1643-dependent dephosphorylation of SMADs indicating that, Kv11.1 is the direct target end mediator of NS1643.”
This statement is not entirely correct, since siRNA did not completely suppressed the NS1643-dependent dephosphorylation of SMADs. Fig. 4 D for Smad 2 seems to lack error bar and statistical significance between NS1643-treated cells in the absence and presence of siRNA against Kv11.1 is not shown. Please correct.
In any case, especially because there is no complete recovery of the phosphorylated state of Smad2/3 even if the channel expression is completely abolished, the authors should at least discuss the points mentioned in my previous review about the specificity issue. For the same reason, to make their findings a strong case, they should repeat the experiments shown in Figures 2A, C and D in one cell line of their choice. If, again, only partial recovery occurs, the authors must clearly point out that NS1643 is exerting its effect also through an additional, Kv11.1-indepenent mechanism.
Finally, the authors did not address my minor points 1 and 2 of the previuos evaluation report, while I believe this would be important in order to give a correct message to the readers in the context of the current knowledge.
Reviewer 2 Report
The manuscript was appropriately revised, and I agree to publish the study in the present state.
Round 2
Reviewer 1 Report
The authos addressed all my concerns and significantly improved the manuscript.
Author Response
We are glad that the reviewer is satisfied with our corrections.
We want to thank the reviewer for his/her time in helping us to improve our work.
This manuscript is a resubmission of an earlier submission. The following is a list of the peer review reports and author responses from that submission.
Round 1
Reviewer 1 Report
In this study, the effects of Kv11.1 (hERG) on the EMT in colon cancer cells are presented. An activator of Kv11.1(NS1643) treatment inhibits EMT and migration via modulating TGFb signaling. The authors showed biochemical evidences, consistent with their interpretation. However, I am not certain whether the effects of NS1643 were actually mediated by K+ channel activity
- The current traces shown in Fig. 1B is hard to understand. Full traces of ionic currents with appropriate scales of current amplitude and time is lacking. The effects on full I/V or G/V curve with original traces are necessary.
- The amplitude (vertical axis) of Fig. 1C is definitely false. Was it really recorded in the colon cancer cell line cells?
- In the schematic figure, the activation of K+ channel (i.e. hyperpolarization) activated Ca2+ influx. Was it actually measured or directly analyzed by using appropriate method? Which kind of Ca2+ channel has been suggested?
- It is definitely necessary to show the inhibitory effect of Kv11.1 blocker on the results induced by NS1643. Also, if the effects shown are induced by hyperpolarization per se, rasing the extracellular K+ concentration by 10 - 20 mM would also inhibit the NS1643 effects. It is necessary to show the K+ channel blocker effects as well as the opener effects.
Reviewer 2 Report
Eskandari et al. Tackles the interesting problem how pharmacological activation of the Kv11.1 channel reprograms EMT in colon cancer cells. The topic itself is very interesting and the group has recently gave important contribution about the role of Kv11.1 in cancer development in various types of cancer. The experiments presented here are well-performed, however some important isssues have to be clarified before publication in order to give a clearcut message.
Major points:
- The major concern is the use of the inhibitors in this study. NS1643 has been shown to activate BK(Ca) channels with an EC50 of 1.8 microM (see https://pubmed.ncbi.nlm.nih.gov/18809671/). This finding is completely ignored in the present manuscript. Here the authors use 50 microM NS1643 in all experiments, meaning that they cannot exclude that at least a part of the effects is due to the action of the drug on BK(Ca) which is equally expressed in the colon cancer lines used in the study (e.g. Wu, H.; Franklin, C.; Kim, H.; Turner, J. J. Physiol. 1991, 260, C35−C42. for HT29 ). The authors write at a certain point: “Also, application of NS1643 with the Kv11.1 blocker E4031 strongly inhibited the effects of NS1643 alone demonstrating that the effects of NS1643 on the TGFβ1 signaling are specific to Kv11.1 activity (Supplementary Fig. 2B)”. This information should be included in the main text, as it is important. On the other hand, please note that E4031 was shown to affect KCa and NCX (https://pubmed.ncbi.nlm.nih.gov/26617732/) . In this study, E4031 was shown to increase reverse-mode of NCX activity, and to trigger preconditioning against infarct size (IS) and arrhythmias caused by ischemia/reperfusion injury through mitoKCa channels. Since the conclusions regarding the role of Kv11.1 in this paper in EMT point to changes in calcium homeostasis, on my opinion the authors should not dismiss the possible effects of the drugs used on BK, mitoKCa and NCX. Thus, it is mandatory to perform the most relevant experiments (e.g. PP2B activation, Smad 3/ phosphorylation, expression of EMT markers) in one of the used cell lines where Kv11.1 expression is downregulated, in the absence and in the presence of NS1643. These experiments would give a definitive answer on the role of Kv11.1 and the specificity of NS1643 used at 50 microM concentration in the processes described here.
Minor points:
- 1 is not found only in the PM. Please cite relevant literature and discuss the possible role of the intracellular Kv11.1 in the observed changes in TGFbeta signalling.
- The authors mention only briefly that several other studies show that instead of activation of Kv11.1, its inhibition was found to decrease tumor volume, even in CRC ( see e.g. https://pubmed.ncbi.nlm.nih.gov/24403225/). The authors must discuss in details these works and try to explain the difference in their results with respect to the results of other groups.
- The Discussion section should be improved. It is mostly the repetition of the findings of the paper instead of a critical discussion (see point 2).
- Also, previous findings by the same group should be discussed, at least the ones describing the effect of the same drug NS1643 on Wnt signalling (https://pubmed.ncbi.nlm.nih.gov/30792401/) and on calcineurin (https://pubmed.ncbi.nlm.nih.gov/25945833/), in the context of the present findings.